**Data Availability Statement:** Data cannot be shared publicly because of patient confidentiality

# Remission, relapse, and risk of major cardiovascular events after metabolic surgery in persons with hypertension: A Swedish nationwide registry-based cohort study

Erik Stenberg[1‡]*, Richard Marsk[2‡], Magnus Sundbom[3], Johan Ottosson[1], Tomas Jernberg[4], Ingmar Näslund[1], Erik Näslund[2]

1 Department of Surgery, Faculty of Medicine and Health, Örebro University, Örebro, Sweden, 2 Division of Surgery, Department of Clinical Sciences, Danderyd Hospital, Karolinska Institutet, Stockholm, Sweden, 3 Department of Surgical Sciences, Uppsala University, Uppsala, Sweden, 4 Division of Cardiovascular Medicine, Department of Clinical Sciences, Danderyd Hospital, Karolinska Institutet, Stockholm, Sweden

‡ These authors share first authorship on this work.
* erik.stenberg@regionorebrolan.se

## Abstract

### Background

Several studies have shown that metabolic surgery is associated with remission of diabetes and hypertension. In terms of diabetes, factors such as duration, insulin use, weight loss, and age have been shown to contribute to the likelihood of remission. Such factors have not been determined for hypertension. The aim of this study was to evaluate factors associated with the remission and relapse of hypertension after metabolic surgery, as well as the risk for major adverse cardiovascular event (MACE) and mortality in patients with and without remission.

### Methods and findings

All adults who underwent metabolic surgery between January 2007 and June 2016 were identified in the nationwide Scandinavian Obesity Surgery Registry (SOReg). Through cross-linkage with the Swedish Prescribed Drug Register, Patient Register, and Statistics Sweden, individual data on prescriptions, inpatient and outpatient diagnoses, and mortality were retrieved. Of the 15,984 patients with pharmacologically treated hypertension, 6,286 (39.3%) were in remission at 2 years. High weight loss and male sex were associated with higher chance of remission, while duration, number of antihypertensive drugs, age, body mass index (BMI), cardiovascular disease, and dyslipidemia were associated with lower chance. After adjustment for age, sex, BMI, comorbidities, and education, the cumulative probabilities of MACEs (2.8% versus 5.7%, adjusted odds ratio (OR) 0.60, 95% confidence interval (CI) 0.47 to 0.77, $p < 0.001$) and all-cause mortality (4.0% versus 8.0%, adjusted OR 0.71, 95% CI 0.57 to 0.88, $p = 0.002$) were lower for patients being in remission at 2 years compared with patients not in remission, despite relapse of hypertension in 2,089 patients (cumulative probability 56.3%) during 10-year follow-up. The main limitations of the

under current Swedish legislation. Data are available from the Scandinavian Obesity Surgery Registry (contact via soreg@regionorebrolan.se), the Swedish Board of Health and Welfare (contact via Registerservice@socialstyrelsen.se) and Statistics Sweden (contact via mikrodata@scb.se) for researchers who meet the criteria for access to confidential data.

**Funding:** ES received grants from Region Örebro County (Grant number: OLL-939106) and the Bengt Ihre Foundation. EN received grants from Stockholm County Council and SRP Diabetes. The funders had no role in study design, data collection and analysis, decision to publish, or preparation of the manuscript.

**Competing interests:** I have read the journal's policy and the authors of this manuscript have the following competing interests: ES has received lecturing fees from Johnson & Johnson Medical. JO has received consultant fees from Vifor Pharma AB, and Johnson & Johnson Medical. IN has received consultant fees from Baricol Bariatrics AB and Johnson & Johnson Medical. None of the mentioned disclosures were related to the contents of this work.

**Abbreviations:** ATC, Anatomical Therapeutic Chemical; BMI, body mass index; CI, confidence interval; EBMIL, excess BMI loss; GATEWAY, Gastric Bypass to Treat Obese Patients With Steady Hypertension; HR, hazard ratio; MACE, major adverse cardiovascular event; NPR, National Patient Register; OR, odds ratio; RCT, randomized controlled trial; RYGB, Roux-en-Y gastric bypass; SD, standard deviation; SG, sleeve gastrectomy; SOReg, Scandinavian Obesity Surgery Registry; SOS, Swedish Obese Subjects; STROBE, Strengthening the Reporting of Observational Studies in Epidemiology; T2D, type 2 diabetes; TWL, total weight loss.

study were missing information on nonpharmacological treatment for hypertension and the observational study design.

## Conclusions

In this study, we observed an association between high postoperative weight loss and male sex with better chance of remission, while we observed a lower chance of remission depending on disease severity and presence of other metabolic comorbidities. Patients who achieved remission had a halved risk of MACE and death compared with those who did not. The results suggest that in patients with severe obesity and hypertension, metabolic surgery should not be delayed.

## Author summary

### Why was this study done?

- Hypertension, particularly in combination with morbid obesity, is a leading cause of mortality and disability worldwide.

- There is a growing body of evidence supporting the reduction of major adverse cardiovascular events (MACEs) and mortality among patients with metabolic comorbidities after bariatric surgery. Less is known of the factors associated with remission and relapse of disease as well as the impact on MACEs and morality from reaching remission.

- The main purpose of this study was to assess which factors that contribute to the remission and relapse of hypertension after metabolic surgery, as well as the risk for MACE and mortality in patients who have achieved remission of hypertension.

### What did the researchers do and find?

- In this nationwide observational study, 15,984 patients with hypertension undergoing a primary metabolic procedure were included.

- Almost 40% of patients with hypertension experienced remission of hypertension 2 years after surgery. While 56% relapsed over 10-year follow-up, those who reached remission experienced lower probability for MACEs and all-cause mortality compared with those who did not reach remission.

### What do these findings mean?

- The results of this study suggest that metabolic surgery has the highest success rate for patients early in the course of disease and thus suggests that metabolic surgery should not be delayed for patients with severe obesity and hypertension.

## Introduction

The association between metabolic surgery and remission of type 2 diabetes (T2D) is well established by several randomized controlled trials (RCTs) and observational studies, demonstrating that metabolic surgery is superior to medical treatment of T2D [1–4]. The association between metabolic surgery and the remission of hypertension is less well studied. In an RCT with remission of hypertension as primary outcome measure, 51% of the surgically treated patients achieved remission of their hypertension, which was superior to medical treatment [5]. In an observational study with 12-year follow-up by Adams and colleagues [6], the surgically treated group had both higher remission rates and lower incidence rates of hypertension than persons with severe obesity not undergoing surgery. A recent meta-analysis concluded that remission rates of hypertension were higher in patients that underwent Roux-en-Y gastric bypass (RYGB) compared with sleeve gastrectomy (SG) [7].

Duration of diabetes, glycemic control, use of insulin, age, and postoperative weight loss have all been shown to be associated with the chance of diabetes remission [1,3,4,8–12]. In terms of hypertension, this has been less studied. The Swedish Obese Subjects (SOS) study reported that the proportion of patients on antihypertensive medications were lower in the surgical group compared with the nonoperated matched controls. They found a significant linear positive relationship between change in blood pressure and change in body mass index (BMI) at 2 years of follow-up but not at 10 years of follow-up [13].

It has recently been shown that metabolic surgery in patients with severe obesity and pharmacologically treated hypertension was associated with lower risk for major adverse cardiovascular events (MACEs) and all-cause mortality compared with age- and sex-matched controls with hypertension from the general population [14]. The impact of postoperative remission of hypertension was not clarified.

The aims of this study were to assess which factors are associated with remission and relapse of hypertension after metabolic surgery as well as to compare the risk of MACE and mortality in patients with and without subsequent remission of their hypertension.

## Methods

This study was conducted using a register linkage based on the Scandinavian Obesity Surgery Registry (SOReg), linked to nationwide Swedish health registers, using the unique personal identity number assigned to each Swedish resident. All data were pseudonymized prior to access.

### Study population and intervention

The SOReg is a national register for bariatric and metabolic surgery, covering more than 97.5% of all bariatric surgery procedures in Sweden [15]. All adults (≥18 years of age) who underwent a primary RYGB or SG at some point between January 1, 2007 and June 30, 2016 with a registration in the SOReg were identified and considered for inclusion. The study used the same study base as that of a recent study [14], with the difference that the present study included all patients with hypertension who underwent a primary gastric bypass or SG procedure in Sweden as registered in the SOReg without matching to controls from the general population.

A cross-linkage was performed with the nationwide Swedish Prescribed Drug Register, the National Patient Register (NPR), the Cause of Death Register, and individual socioeconomic data from Statistics Sweden. The Swedish Prescribed Drug Register was established in 2005 and includes all dispensed prescription drugs (updated monthly) classified according to the

World Health Organization Anatomical Therapeutic Chemical (ATC) classification system. The Drug Register was complete up until December 31, 2018 at time of retrieval. The NPR is a nationwide register to which hospitals are obliged to report all inpatient and outpatient hospital care data [16]. The inpatient component of the register attained national coverage in 1987 and covers nearly 100% of all hospital admissions in public healthcare, while the outpatient component, started in 2001, has successively reached about 96% of outpatient visits in specialized healthcare. Data from the NPR were available until December 31, 2017. The Total Population Register, continually updated by Statistics Sweden, provides data on emigration/immigration, marriage/divorce, and complete coverage of dates of birth/death for each individual in Sweden [17]. The Education Register from Statistics Sweden provided data on educational levels at the time of surgery. Perioperative data, information on postoperative complications, and weight loss were based on data from the SOReg.

## Inclusion and exclusion criteria

There were no mandatory national eligibility criteria for bariatric surgery during the study period, although most regions in Sweden considered BMI $\geq$35 kg/m$^2$ with or without obesity-related comorbidity as eligibility criteria. Hypertension was defined by the use of antihypertensive drugs (ATC codes C02, C03, C07, C08, or C09) within 18 months prior to surgery. As some of these drugs also are used in treating heart failure, atrial fibrillation, and tachycardia, individuals with the following combinations of diagnoses and prescriptions were excluded: heart failure (ICD-10: I50) or cardiomyopathy (ICD-10: I42) and C03C, C07A, C09A, C09B, or C09C; atrial fibrillation, flutter, or other tachycardia (ICD-10: I47 and I48) and C07 or C08D. Patients without hypertension at the time of surgery were also excluded (Fig 1).

## Covariates

Educational level was retrieved from the Education Register at Statistics Sweden. Dispensed prescription drugs were retrieved from the Prescribed Drug Register. Baseline BMI and the presence of sleep apnea, depression, and diabetes were based on data from the SOReg and defined as active treatment (continuous positive airway pressure and pharmacological treatment, respectively) at the time of surgery. The definitions of dyslipidemia and cardiovascular comorbidity were based on data from the Swedish Prescribed Drug Register. Dyslipidemia was defined by the use of lipid modifying drugs (ATC code C10). Diabetes was defined as being

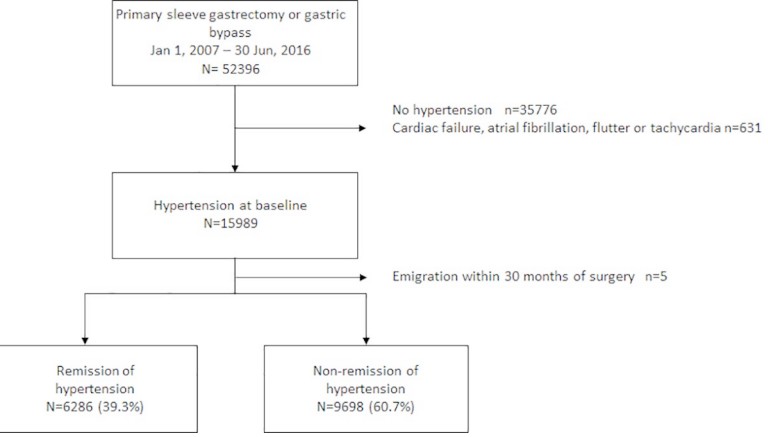

**Fig 1. Flowchart describing participant's identification and inclusion.**

prescribed antidiabetic drugs (ATC code A10). Cardiovascular comorbidity was defined as a previous diagnose of heart failure (ICD-10: I50), acute myocardial infarction or angina pectoris (ICD-10: I20 to I22), atrial fibrillation, flutter, or other tachycardia (ICD-10: I47 and I48) not excluded based on the exclusion criteria as listed above. Cerebrovascular disease was defined as a previous cerebrovascular event (ICD-10: I60 to I64).

## Procedures

The surgical technique for the laparoscopic gastric bypass procedure is highly standardized in Sweden with the majority being an antecolic, antegastric RYGB with a small gastric pouch (<25 mL), an alimentary limb of 100 cm, and a biliopancreatic limb of 50 cm [18]. The surgical technique for the laparoscopic SG is less standardized, but routinely performed using a 32–36 Fr bougie, starting the resection no more than 5 cm from the pylorus, ending the resection 1 cm from the angle of His.

## Outcome and follow-up

The main outcome was the remission of pharmacological treatment of hypertension defined as no prescribed antihypertensive drugs for 1 entire year (18 to 30 months after surgery). Patients who died during the first 30 months after surgery were included in the analyses on remission (and considered to not reach remission), but based on this definition, they were excluded from the analyses on mortality, MACE, and relapse since their inclusion would likely overestimate the impact of remission.

Secondary outcomes were all-cause mortality and MACEs, defined as the first occurrence of unstable angina, myocardial infarction, acute cerebrovascular event, fatal cardiovascular event (cause of death ICD-10: I01 to I78, excluding I30), or unattended sudden cardiac death (ICD-10: R96.0, R96.1, R98, and R99). Any adverse event during the first 30 days after surgery were considered to be a postoperative complication. The Clavien–Dindo classification of postoperative complications was introduced in the SOReg on January 1, 2010 [19]. Complications graded as ≥3b (i.e., a complication requiring intervention under general anesthesia, resulting in organ failure or death) were considered to be serious complications. A tertiary outcome was improvement in hypertension estimated as a reduction of at least 30% of the total number of antihypertensive medications [20].

## Statistics

Postoperative weight loss is presented as change in BMI (BMI loss = initial BMI − postoperative BMI), percentage total weight loss (TWL = 100 × weight loss/preoperative weight), and percentage excess BMI loss (EBMIL = 100 × [initial BMI–postoperative BMI]/[initial BMI– 25]). Categorical data are presented as numbers ($n$) and percentage (%), continuous variables as mean +/− standard deviation (SD). Differences in weight loss depending on surgical method was evaluated using the Student $t$ test. Univariable logistic regression and multivariable logistic regression (including age, BMI, duration of antihypertensive treatment (more recent or longer than 1 year), and severity of hypertension (number antihypertensive drug prescribed at baseline), age, sex, comorbidities (dyslipidemia, sleep apnea, depression, diabetes, and cardiovascular and cerebrovascular comorbidities), and TWL at 1 year after surgery) were used to assess the chance of reaching remission, reporting odds ratios (ORs) with 95% confidence intervals (CIs) as measures of association. Risk for relapse of hypertension was estimated using univariable and multivariable Cox regression (including the same variables as listed above). Time to relapse and first episodes of MACE and all-cause mortality were estimated and visualized using the Kaplan–Meier method. Comparisons were made by the log-rank test. Risks for

MACE and mortality were estimated using univariable and multivariable Cox regression including factors potentially affecting the risk for these outcomes (adjusted for age, sex, BMI, dyslipidemia, depression, sleep apnea, diabetes, cardiovascular and cerebrovascular comorbidities, and education). The regression models were tested for multicollinearity using linear regression with a variance inflation factor >5 considered to indicate an issue with multicollinearity. No multicollinearity issues were detected in either of the multivariable models. Participants were followed from the surgery date until emigration, death, or end of follow-up, whichever came first. The Wilcoxon signed rank test was used to evaluate differences in numbers of antihypertensive medications.

Although an original study plan was decided on by the authors, it was not officially documented beforehand. A post hoc analysis (not within the original study plan) in the form of secondary analyses on remission and relapse were made stratified by sex, while weight loss results were stratified by surgical method. In addition, the numbers of antihypertensive drugs before and after surgery were presented in numbers in S3 Table. A tertiary analysis concerning improvement of hypertension was included to allow comparison to other important recent studies in the field. Finally, the Bonferroni–Holm method was included to adjust for multiple calculations separately in all of the multivariable analyses [21].

IBM SPSS version 25 (IBM, Armonk, New York, United States of America) was used for all statistical analyses.

### Ethics

The study was approved by the regional ethical review board of Stockholm (ref nr 2017/857-32). No written consent was obtained from the study participants. However, in accordance with Swedish legislation, all participants were informed of the research and quality registry and that the data will be used in clinical research, giving the patients the right to deny participation.

This study is reported as per the Strengthening the Reporting of Observational Studies in Epidemiology (STROBE) guideline (S1 Checklist).

## Results

During the study period, 52,396 patients were identified. After exclusion of patients with heart failure, atrial fibrillation and tachycardia, and patients without hypertension at baseline, 15,989 patients were included in the study cohort. After exclusion of 5 patients who emigrated within 30 months of surgery, 15,984 remained in the study (Fig 1).

Follow-up for 30-day complications was 98.7% ($n = 15,778$). The follow-up rate for weight loss was 89.7% ($n = 14,332$) at 1 year and 59.8% ($n = 9,559$) at 2 years after surgery. All patients were followed for mortality and the use of prescribed drugs until the end of the study. Mean follow-up time was $6.5 \pm 2.3$ years. Baseline characteristics for the study group is presented in Table 1.

### Surgery and weight outcome

The majority of the surgical procedures were laparoscopic 15,365 (96.1%), with conversion to open surgery in 172 operations (1.1%), while the remaining 447 (2.8%) procedures were primarily performed using an open technique. A postoperative complication of any type occurred for 1,531 patients (9.7%) within 30 days after surgery, and of the 13,403 patients operated after January 1, 2010 with a follow-up at day 30, 472 (3.5%) experienced a serious postoperative complication. Mean BMI loss 1 year after surgery was $12.6 \pm 4.0$ kg/m$^2$, with TWL $29.6 \pm 7.8$% and EBMIL $76.2 \pm 23.0$%. Mean BMI loss 2 years after surgery was $12.7 \pm 4.6$ kg/m$^2$, with

**Table 1. Baseline cha racteristics of the study cohort.**

| | Missing data | Entire cohort | Remission at 2 years | Nonremission at 2 years | p-Value[1] |
|---|---|---|---|---|---|
| Total number of patients | | 15,984 | 6,286 (39.3%) | 9,698 (60.7%) | |
| Hypertension duration >1 year, n (%) | 0 (0.0%) | 13,268 (83.0%) | 4,406 (33.2%) | 8,862 (66.8%) | <0.001 |
| Numbers of preoperative drugs, n (%) | 0 (0.0%) | | | | |
| 1 | | 5,864 (36.7%) | 3,624 (61.8%) | 2,240 (38.2%) | Reference |
| 2 | | 4,524 (28.3%) | 1,650 (36.5%) | 2,874 (63.5%) | <0.001 |
| 3 | | 3,149 (19.7%) | 708 (22.5%) | 2,441 (77.5%) | <0.001 |
| 4 | | 1,646 (10.3%) | 240 (14.6%) | 1,406 (85.4%) | <0.001 |
| ≥5 | | 801 (5.0%) | 64 (8.0%) | 737 (92.0%) | <0.001 |
| Age, mean ± SD | 0 (0.0%) | 48.7 ± 9.0 | 45.5 ± 9.2 | 50.8 ± 8.2 | <0.001 |
| BMI, mean ± SD | 0 (0.0%) | 42.2 ± 5.6 | 42.2 ± 5.7 | 42.3 ± 5.6 | 0.505 |
| Sex, n (%) | 0 (0.0%) | | | | |
| Female | | 10,608 (66.4%) | 4,338 (40.9%) | 6,270 (59.1%) | Reference |
| Male | | 5,376 (33.6%) | 1,948 (36.2%) | 3,428 (63.8%) | <0.001 |
| Comorbid disease, n (%) | 0 (0.0%) | | | | |
| Dyslipidemia | | 5,208 (32.6%) | 1,395 (26.8%) | 3,813 (73.2%) | <0.001 |
| Depression | | 2,694 (16.9%) | 1,028 (38.2%) | 1,666 (61.8%) | 0.174 |
| Sleep apnea | | 2,732 (17.1%) | 901 (33.0%) | 1,831 (67.0%) | <0.001 |
| Diabetes | | 4,989 (31.2%) | 1,580 (31.7%) | 3,409 (68.3%) | <0.001 |
| Cardiovascular comorbidity | | 970 (6.1%) | 125 (12.9%) | 845 (87.1%) | <0.001 |
| Cerebrovascular disease | | 316 (2.0%) | 64 (20.3%) | 252 (79.7%) | <0.001 |
| Education, n (%) | 59 (0.4%) | | | | |
| Primary education | | 2,896 (18.2%) | 1,082 (37.4%) | 1,814 (62.6%) | 0.022 |
| Secondary education | | 9,419 (59.1%) | 3,743 (39.7%) | 5,676 (60.3%) | Reference |
| Higher education | | 3,610 (22.7%) | 1,435 (39.8%) | 2,175 (60.2%) | 0.990 |
| Surgical method, n (%) | 0 (0.0%) | | | | |
| Gastric bypass | | 14,821 (92.7%) | 5,886 (39.7%) | 8,935 (60.3%) | Reference |
| SG | | 1,163 (7.3%) | 400 (34.4%) | 763 (65.6%) | <0.001 |

[1] p-Value based on unadjusted logistic regression.

BMI, body mass index (in kg/m$^2$); n, numbers; SD, standard deviation; SG, sleeve gastrectomy.

TWL 29.9± 9.1% and EBMIL 76.7 ± 24.8%. When stratified by surgical method, the mean BMI loss 1 year after surgery was 12.8 ± 3.9 kg/m$^2$ versus 9.9 ± 4.2 kg/m$^2$, with TWL 30.0 ± 7.6% versus 24.2 ± 8.7%, and EBMIL 76.8 ± 22.5% versus 67.5 ± 27.1% for gastric bypass and SG, respectively ($p < 0.001$). Mean BMI loss at 2 years after surgery was 12.9 ± 4.5 kg/m$^2$ versus 9.1 ± 4.7 kg/m$^2$, TWL 30.3 ± 8.9% versus 22.6 ± 10.1%, and EBMIL 77.3 ± 24.3% versus 63.3 ± 30.7%, respectively ($p < 0.001$).

## Remission of pharmacological treatment of hypertension

At 2 years after surgery, 6,286 patients (39.3%) had been taken off their pharmacological treatment for hypertension. The duration of disease, number of antihypertensive drugs, age, BMI, dyslipidemia, and cardiovascular disease were all associated with a lower remission rate, while higher postoperative weight loss and male sex were associated with a higher chance of reaching remission (Table 2, S1 Table).

The number of antihypertensive medications was reduced from a median of 2 (IQR 1 to 3) before surgery to a median of 1 (IQR 0 to 2) after surgery ($p < 0.001$). A reduction of at least

**Table 2. Chance of reaching hypertension remission 2 years after surgery.**

| | Unadjusted OR | Adjusted OR | Adjusted *p*-value[1] |
|---|---|---|---|
| Hypertension duration >1 year | 0.22 (0.20 to 0.24) | 0.41 (0.37 to 0.46) | <0.001* |
| Numbers of preoperative drugs | | | |
| 1 | Reference | Reference | Reference |
| 2 | 0.35 (0.33 to 0.38) | 0.45 (0.41 to 0.50) | <0.001* |
| 3 | 0.18 (0.16 to 0.20) | 0.26 (0.23 to 0.29) | <0.001* |
| 4 | 0.11 (0.09 to 0.12) | 0.16 (0.13 to 0.18) | <0.001* |
| ≥5 | 0.05 (0.04 to 0.07) | 0.09 (0.07 to 0.12) | <0.001* |
| %TWL, 1 year after surgery | 1.05 (1.04 to 1.05) | 1.04 (1.03 to 1.04) | <0.001* |
| Age | 0.93 (0.93 to 0.94) | 0.96 (0.96 to 0.97) | <0.001* |
| BMI | 1.00 (0.99 to 1.00) | 0.98 (0.97 to 0.98) | <0.001* |
| Sex | | | |
| Female | Reference | Reference | Reference |
| Male | 0.82 (0.77 to 0.88) | 1.20 (1.10 to 1.32) | <0.001* |
| Comorbid disease | | | |
| Dyslipidemia | 0.44 (0.41 to 0.47) | 0.87 (0.78 to 0.95) | 0.004* |
| Depression | 0.94 (0.87 to 1.03) | 0.93 (0.84 to 1.03) | 0.186 |
| Sleep apnea | 0.72 (0.66 to 0.78) | 1.00 (0.89 to 1.11) | 0.932 |
| Diabetes | 0.59 (0.55 to 0.64) | 1.02 (0.93 to 1.13) | 0.625 |
| Cardiovascular comorbidity | 0.21 (0.18 to 0.26) | 0.45 (0.36 to 0.56) | <0.001* |
| Cerebrovascular disease | 0.39 (0.29 to 0.51) | 1.03 (0.74 to 1.45) | 0.850 |
| Education | | | |
| Primary education | 0.90 (0.83 to 0.99) | 1.05 (0.95 to 1.17) | 0.323 |
| Secondary education | Reference | Reference | Reference |
| Higher education | 1.00 (0.93 to 1.08) | 1.02 (0.93 to 1.12) | 0.633 |
| Surgical method | | | |
| Gastric bypass | Reference | Reference | Reference |
| SG | 0.80 (0.70 to 0.90) | 0.87 (0.74 to 1.01) | 0.075 |

[1] Multivariable, logistic regression model, including all variables in the table.

* Significant value after correction with the Bonferroni–Holm method.

%TWL, percentage total weight loss; BMI, body mass index; CI, confidence interval; OR, odds ratio (presented with 95% CI); *n*, numbers; SG, sleeve gastrectomy.

30% of the numbers of antihypertensive drugs was seen in 10,290 patients (64.9%) at 2 years after surgery (S3 Table, S1 Fig).

## Relapse of hypertension

Of the 6,286 patients who initially reached remission, relapse occurred in 2,089 patients within 10 years (cumulative probability 56.3 ± 2.4%) (Fig 2). The duration of disease, number of antihypertensive drugs, age, and SG were all associated with a higher risk for relapse, while higher postoperative weight loss was associated with a lower risk for relapse (Table 3, S2 Table).

## MACEs

At 8 years after surgery, MACEs had occurred for 90 patients who initially reached remission (cumulative probability 2.8 ± 0.6%), compared with 366 patients who did not reach remission (cumulative probability 5.7 ± 0.8%): unadjusted hazard ratio (HR) 0.42 (95% CI 0.33 to 0.53, $p < 0.001$) and adjusted HR 0.60 (95% CI 0.47 to 0.77, adjusted $p < 0.001$) (Fig 3, S4 Table).

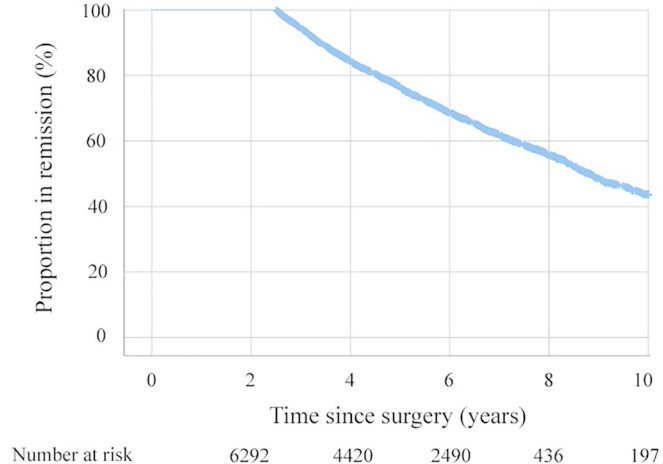

**Fig 2. Relapse-free survival for patients experiencing remission of hypertension at 2 years after metabolic surgery (unadjusted Kaplan–Meier curve).**

## Mortality

At 10 years after surgery, 115 patients who initially reached remission had died (cumulative probability 4.0 ± 1.0%), compared with 345 patients who did not reach remission (cumulative probability 8.0 ± 1.2%): unadjusted HR 0.51 (95% CI 0.41 to 0.62, $p < 0.001$) and adjusted HR 0.71 (95% CI 0.57 to 0.88, adjusted $p = 0.002$) (Fig 4, S5 Table).

## Discussion

This study has demonstrated that disease severity, duration, degree of weight loss, age, sex, and metabolic comorbidities are all factors associated with the chance of remission of hypertension after metabolic surgery. The risk of relapse is associated with severity of disease, duration, degree of weight loss, age as well as surgical procedure. Patients who achieve remission of their hypertension 2 years after metabolic surgery have a lower risk of MACE and mortality than those who do not.

There is a high degree of similarity between factors associated with the remission of T2D and the factors found to be associated with the remission of hypertension in this study [4]. Both are chronically progressive diseases with obesity as a shared risk factor [22]. Similar to the effects of metabolic surgery on T2D [23], the effect on blood pressure is seen as early as in the first week after surgery [24], thus suggesting that the cardiovascular and metabolic improvements may at least partially be mediated by similar mechanisms. Although the degree of weight loss was found to be associated with the chance of remission of hypertension in this study, as earlier shown for T2D [4], the effect on both hypertension and T2D occurs before any significant weight loss has occurred. This, and the recent observation that the risk of MACE decreases after approximately 10% of weight loss in bariatric surgical patients with T2D (of which 82% had hypertension) compared with approximately 20% in a matched nonsurgical group [25], suggests that other factors than just weight loss might be in play. Voluntary, nonsurgical weight loss has previously been associated with increased diuresis and negative potassium and sodium balance during the initial weight loss phase [26]. An increased diuresis after longer follow-up time has also been described after gastric bypass, with a linear association between daily urinary output and effects on blood pressure [13].

**Table 3. Factors associated with risk for relapse of hypertension.**

| | Unadjusted HR | Adjusted HR | Adjusted p-value[1] |
|---|---|---|---|
| Hypertension duration >1 year | 1.69 (1.53 to 1.88) | 1.46 (1.30 to 1.63) | <0.001* |
| Numbers of preoperative drugs | | | |
| 1 | Reference | Reference | Reference |
| 2 | 1.58 (1.43 to 1.75) | 1.39 (1.25 to 1.55) | <0.001* |
| 3 | 2.20 (1.95 to 2.48) | 1.83 (1.59 to 2.09) | <0.001* |
| 4 | 2.53 (2.11 to 3.04) | 2.33 (1.92 to 2.84) | <0.001* |
| ≥5 | 3.61 (2.63 to 4.96) | 2.99 (2.11 to 4.25) | <0.001* |
| %TWL 1 year after surgery | 0.98 (0.98 to 0.99) | 0.98 (0.98 to 0.99) | <0.001* |
| Age | 1.02 (1.02 to 1.03) | 1.01 (1.01 to 1.02) | <0.001* |
| BMI | 1.00 (0.99 to 1.00) | 1.01 (1.00 to 1.02) | 0.050 |
| Sex | | | |
| Female | Reference | Reference | Reference |
| Male | 1.14 (1.04 to 1.24) | 1.00 (0.90 to 1.11) | 0.992 |
| Comorbid disease | | | |
| Dyslipidemia | 1.23 (1.12 to 1.36) | 0.96 (0.85 to 1.08) | 0.475 |
| Depression | 1.08 (0.96 to 1.21) | 1.06 (0.93 to 1.20) | 0.364 |
| Sleep apnea | 1.17 (1.04 to 1.31) | 1.04 (0.91 to 1.18) | 0.573 |
| T2D | 1.17 (1.06 to 1.28) | 1.05 (0.94 to 1.18) | 0.379 |
| Cardiovascular comorbidity | 1.55 (1.19 to 2.03) | 1.07 (0.78 to 1.48) | 0.663 |
| Cerebrovascular disease | 1.53 (1.06 to 2.19) | 1.24 (0.81 to 1.90) | 0.328 |
| Education | | | |
| Primary education | 1.12 (1.00 to 1.25) | 1.08 (0.95 to 1.22) | 0.245 |
| Secondary education | Reference | Reference | Reference |
| Higher education | 0.94 (0.89 to 1.11) | 0.95 (0.85 to 1.07) | 0.418 |
| Surgical method | | | |
| Gastric bypass | Reference | Reference | Reference |
| SG | 1.47 (1.17 to 1.85) | 1.48 (1.16 to 1.90) | 0.002* |

[1] Multivariable Cox regression model, including all variables in the table.

* Significant value after correction with the Bonferroni–Holm method.

%TWL, percentage total weight loss; BMI, body mass index; CI, confidence interval; HR, hazard ratio (presented with 95% CI); n, numbers; SG, sleeve gastrectomy; T2D, type 2 diabetes.

We found a 2-year remission rate of hypertension of 39%. This is lower than the 63% resolution or improvement of hypertension reported by Vest and colleagues after a mean follow-up of 5 years in a systematic review [27], but similar to the remission rates reported in the Gastric Bypass to Treat Obese Patients With Steady Hypertension (GATEWAY) trial [20] as well as those reported in a recent Norwegian cohort study [28]. Although it is possible that some patients in our study were maintained on the prescribed medication despite having achieved remission, even patients not reaching remission may well be able to reduce the number of drugs needed to reach sufficient blood pressure levels [14,20]. Using a similar definition of improvement to that of the GATEWAY trial, 65% of the patients in our study were able to reduce their number of antihypertensive medications by at least 30%. This rate compares well to that reported in previous studies [20,27].

The true effect of metabolic surgery on hypertension has been questioned [29], and high rates of relapse have been reported. We found that 56% of those who went into initial remission of hypertension eventually relapsed at 10 years after surgery. Again, the duration and

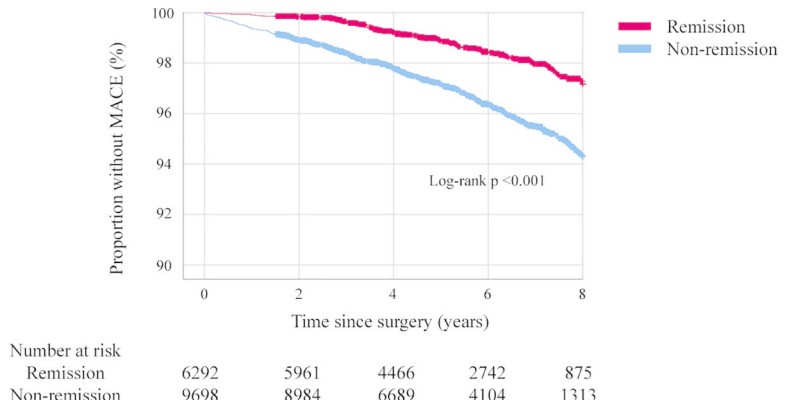

**Fig 3. MACE-free survival in patients experiencing remission of hypertension at 2 years and those not experiencing remission (unadjusted Kaplan–Meier curve).** MACE, major adverse cardiovascular event.

severity of disease were the main factors associated with the risk for relapse. In contrast to remission, the SG was associated with higher risk of relapse even after adjustment for other potential explanatory factors such as postoperative weight loss. Although this finding should be viewed with some caution with respect to the nonrandomized design, it supports the growing body of evidence supporting a better metabolic effect of the gastric bypass procedure that goes beyond strict weight loss effects [30,31]. In the SOS study, the use of antihypertensive medications was reduced from 39% to 27% at 2-year follow-up after RYGB, while no difference was seen at 10 years, suggesting a significant relapse of hypertension. Furthermore, the linear association between the reduction in blood pressure and weight loss at 2 years was not evident at 10-year follow-up. The number of patients who underwent RYGB were, however, few in the SOS study, and only 68% were followed for 10 years [13]. In the present study, the follow-up rate for the pharmacological treatment of hypertension was 100%.

Although the risk for relapse of hypertension is high, patients who went into initial remission also experienced significant reduction in the risk for MACE as well as all-cause mortality. Several recent reports have highlighted a reduced risk of MACE and mortality in patients with T2D or hypertension that undergo metabolic surgery. Aminian and colleagues [31] recently

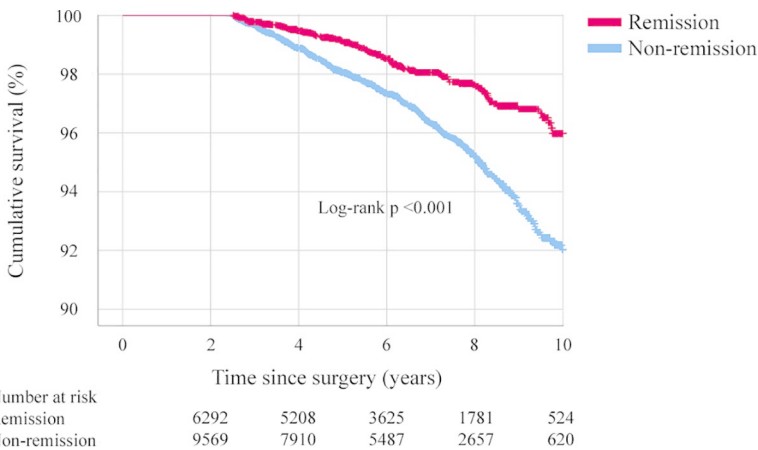

**Fig 4. Overall survival in patients experiencing remission of hypertension at 2 years and those not experiencing remission (unadjusted Kaplan–Meier curve).**

reported a significant difference in favor of metabolic surgery for an extended MACE outcome (adjusted HR 0.61; 95% CI [0.55 to 0.69]) in patients with obesity and T2D. Stenberg and colleagues [14] reported a reduction in risk of MACE after metabolic surgery (adjusted HR 0.73, 95% CI 0.64 to 0.84, $p < 0.001$) in 11,863 patients with obesity and pharmacologically treated hypertension compared with 26,199 matched individuals with hypertension from the general population, after adjustment for duration of hypertension, comorbidities, and education. In light of this, our finding that the risk for MACE and mortality was lower in those who achieved remission of hypertension is not surprising, but it highlights the importance of prioritizing patients having much to gain from metabolic surgery. Further studies focused on optimizing the results for patients who do not reach remission is also needed.

Despite the strengths of the large, nationwide study population and the use of high-quality data from several sources, this study has several limitations. First, this is an observational study. Thus, we cannot exclude that residual confounding factors exist. We, therefore, need to be cautious regarding causality. The study group was limited to patients receiving pharmacological treatment for hypertension, thus excluding patients with undiagnosed hypertension before surgery as well as patients with low adherence to treatment. The results of this study may not be applicable to these specific situations, which deserves the focus of future studies. Furthermore, data on the remission of hypertension after metabolic surgery were based on the discontinuation of pharmacological treatment that may result in an overestimation as well as underestimation of the true remission rates. We, however, believe that underestimation is more likely given the inertia of the healthcare system when it comes to discontinuation of chronic medication. As with all registry studies, there may exist coding errors. On the other hand, this study was based on large nationwide registers with known high validity and degree of completeness.

## Conclusions

This study has demonstrated that metabolic surgery is associated with a significant degree of remission of hypertension with a high postoperative weight loss and male sex being associated with better chance of remission, while age, disease severity, and presence of other metabolic comorbidities being associated with lower chance. Although more than half of the patients relapse within 10 years after surgery, the risks for MACE and mortality are markedly reduced compared with those who do not achieve remission at 2 years. Taken together, this suggests that in patients with severe obesity and hypertension, metabolic surgery should not be delayed.

## Supporting information

**S1 Checklist. STROBE Checklist.** STROBE, Strengthening the Reporting of Observational Studies in Epidemiology.
(DOCX)

**S1 Table. Multivariable logistic regression model for the chance of reaching hypertension remission at 2 years after surgery stratified by sex.**
(DOCX)

**S2 Table. Multivariable Cox regression model for factors associated with risk for relapse of hypertension stratified by sex.**
(DOCX)

**S3 Table. Numbers of antihypertensive drugs before and 2 years after surgery.**
(DOCX)

**S4 Table. Multivariable Cox regression model for the occurrence of MACE up to 8 years after surgery.** MACE, major adverse cardiovascular event.
(DOCX)

**S5 Table. Multivariable Cox regression model for the occurrence of mortality up to 10 years after surgery.**
(DOCX)

**S1 Fig. Stacked histogram of numbers of antihypertensive drugs before and 2 years after surgery.**
(TIF)

## Author Contributions

**Conceptualization:** Erik Stenberg, Richard Marsk, Magnus Sundbom, Johan Ottosson, Tomas Jernberg, Ingmar Näslund, Erik Näslund.

**Data curation:** Erik Stenberg, Richard Marsk, Johan Ottosson, Ingmar Näslund.

**Formal analysis:** Erik Stenberg, Richard Marsk.

**Funding acquisition:** Erik Stenberg, Erik Näslund.

**Investigation:** Erik Stenberg, Richard Marsk, Magnus Sundbom, Ingmar Näslund, Erik Näslund.

**Methodology:** Erik Stenberg, Richard Marsk, Magnus Sundbom, Tomas Jernberg, Erik Näslund.

**Project administration:** Johan Ottosson, Ingmar Näslund, Erik Näslund.

**Resources:** Magnus Sundbom, Erik Näslund.

**Software:** Erik Stenberg, Richard Marsk, Johan Ottosson.

**Supervision:** Tomas Jernberg, Erik Näslund.

**Validation:** Erik Stenberg.

**Visualization:** Erik Stenberg.

**Writing – original draft:** Erik Stenberg, Richard Marsk.

**Writing – review & editing:** Magnus Sundbom, Johan Ottosson, Tomas Jernberg, Ingmar Näslund, Erik Näslund.

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
