## [Editor Report · Decision Letter 0]

11 Mar 2021

Dear Dr Stenberg, 

Thank you for submitting your manuscript entitled "Remission, relapse and risk of major cardiovascular events after metabolic surgery in subjects with hypertension – a nationwide cohort study" for consideration by PLOS Medicine.

Your manuscript has now been evaluated by the PLOS Medicine editorial staff and I am writing to let you know that we would like to send your submission out for external peer review.

Please re-submit your manuscript within two working days, i.e. by March 15, 2021.

Kind regards,

Beryne Odeny

Associate Editor

PLOS Medicine

---

## [Decision Letter · Decision Letter 1]

14 May 2021

Dear Dr. Stenberg,

Thank you very much for submitting your manuscript "Remission, relapse and risk of major cardiovascular events after metabolic surgery in subjects with hypertension – a nationwide cohort study" (PMEDICINE-D-21-01171R1) for consideration at PLOS Medicine. 

[LINK]

In light of these reviews, I am afraid that we will not be able to accept the manuscript for publication in the journal in its current form, but we would like to consider a revised version that addresses the reviewers' and editors' comments. Obviously we cannot make any decision about publication until we have seen the revised manuscript and your response, and we plan to seek re-review by one or more of the reviewers. 

We expect to receive your revised manuscript by Jun 04 2021 11:59PM. Please email us (plosmedicine@plos.org) if you have any questions or concerns.

We look forward to receiving your revised manuscript. 

Sincerely,

Beryne Odeny, 

PLOS Medicine

plosmedicine.org

1) Please revise your title according to PLOS Medicine's style. Your title must be nondeclarative and not a question. It should begin with main concept if possible. Please include the country/ setting and place the study design ("A retrospective cohort study,") in the subtitle (i.e., after a colon). 

2) The Data Availability Statement (DAS) requires revision. For each data source used in your study: If the data are not freely available, please describe briefly the ethical, legal, or contractual restriction that prevents you from sharing it. Please also include an appropriate contact (web or email address) for inquiries (again, this cannot be a study author/ co-author).

3) Abstract:

a) Please ensure that all numbers presented in the abstract are present and identical to numbers presented in the main manuscript text.

b) Please include the important dependent variables that are adjusted for in the analyses.

c) In the last sentence of the Abstract Methods and Findings section, please describe the main limitation(s) of the study's methodology.

d) Abstract Conclusions: Please address the study implications without overreaching what can be concluded from the data; the phrase "In this study, we observed ..." may be useful. Please interpret the study based on the results presented in the abstract, emphasizing what is new.

4) Did your study have a prospective protocol or analysis plan? Please state this (either way) early in the Methods section. 

5) Please state whether the data were de-identified (anonymized) prior to access.

6) Please add the following statement, or similar, to the Methods: "This study is reported as per the Strengthening the Reporting of Observational Studies in Epidemiology (STROBE) guideline (S1 Checklist)." 

7) Thank you for providing your STROBE checklist. Please replace the page numbers with paragraph numbers per section (e.g. "Methods, paragraph 1"), since the page numbers of the final published paper may be different from the page numbers in the current manuscript.

8) Please remove language that implies causality, such as “influence” or “results in” in the discussion and conclusion sections. Refer to associations instead.

9) Please describe how you selected your adjustment variables.

10) We note the potential for unobservable confounding in this observational study. Have you considered using robust methods such as propensity score matching or instrumental variables to address this. Please discuss.

11) In your statistical analyses, please use hierarchical/ multilevel models given that nationwide data is likely clustered at various regional levels. The potential clustering of data (e.g., among patients from the same locality or hospital) would result in spurious effect estimates and standard errors

12) In statistical methods, please refer to any post-hoc corrections to correct for multiple comparisons during your statistical analyses. If these were not performed please justify the reasons. Please refer to our statistical reporting guidelines for assistance (https://journals.plos.org/plosone/s/submission-guidelines.#loc-statistical-reporting)

13) Please provide 95% CIs and p values for estimates in the main text and tables

14) Please specify the statistical test used to derive a p value.

15) Please define the abbreviations in tables and/or Figures e.g. BMI, HR etc.

16) Please replace "subject" with participant, patient, individual, or person.

17) The terms gender and sex are not interchangeable (as discussed in http://www.who.int/gender/whatisgender/en/ ); please use the appropriate term.

Comments from the reviewers:

Reviewer #1: See attachment

Michael Dewey

Reviewer #2: Dear authors,

This is a very interesting and well written manuscript.

I have a few comments about it:

Is this cohort the same of your previous study? The previous publication presented 17531 patients with hypertension. Can you explain the difference in the numbers?

Why did you exclude patients who died during the first 30 months from the analyses on

Mortality and MACE?

As you excluded patients with HF, atrial fibrillation and tachycardia, what is considered cardiovascular comorbidity.

Table 1 needs revision because:

1. It should show patients with hypertension duration<1year to be clearer.

2. The % should totalize 100% in the lines, not in the columns. For example, it is better to know the percentage of remission and non-remission in patients using 1 drug.

3. The use of 1 drug is considered the reference, but it is not explained in the legends.

Reviewer #3: The authors address an appropriate and pertinent question regarding the relationship of remission of hypertension following bariatric surgery to various outcomes. Several national databases are used to extract the data. This reviewer has the following questions or comments.

As opposed to other studies utilizing the SOS database, this study is not a comparison of untreated controls with surgical patients. It is not clear in the abstract that the study is in fact a comparison of surgical patients who did or did not experience remission of hypertension at two years.

In the methods, it is stated that deaths in the first 30 months following surgery are excluded. Given that the primary outcomes are reported at 24 months, it is not clear why this exclusion has been done.

Multiple databases have been used to establish the various metrics reported. It is not clear to this reviewer what the differences between the SOReg and SOS are and whether this creates an inconsistency in the results.

While the definition of hypertension in this study in participants requiring medical treatment/pharmacologic treatment of hypertension is clear, the likelihood of the presence of hypertension in this population at baseline despite a lack of pharmacologic intervention may be important. A similar phenomenon may occur among these participants in the follow up.

In the results, it is unclear why the weight loss data was available in just 60% of participants when all other data was apparently available in nearly 100%. If the mortality data for example was based on a different database, it is reasonable to assume it was 100%. This should be clarified.

In tables 1 and 2, data for bypass versus a sleeve are included but little is said about this comparison in the body of the manuscript. Given the clinical interest in this issue, more could be said regarding this issue. The weight loss for example is reported merging these two populations into a single population.

In the discussion, the authors hypothesize that the mechanisms of early remission of T2DM and hypertension are similar. This reviewer suggests caution regarding this statement as no direct data are reported. It is known that in the early phase of acute weight loss, there is a diuresis of sodium and water which may well be responsible for the early response regarding hypertension but not diabetes. It is not clear to this reviewer why in figure 2, the portion in remission is depicted as 100% at 2+ years when the results report 39%.

[LINK]

---

## [Decision Letter · Decision Letter 2]

26 Jul 2021

Dear Dr. Stenberg,

Thank you very much for re-submitting your manuscript "Remission, relapse and risk of major cardiovascular events after metabolic surgery in persons with hypertension: a Swedish nationwide registry-based cohort study" (PMEDICINE-D-21-01171R2) for review by PLOS Medicine.

I have discussed the paper with my colleagues and the academic editor and it was also seen again by three reviewers. I am pleased to say that provided the remaining editorial and production issues are dealt with we are planning to accept the paper for publication in the journal.

[LINK]

We look forward to receiving the revised manuscript by Jul 26 2021 11:59PM.   

Sincerely,

Beryne Odeny, 

Associate Editor 

PLOS Medicine

plosmedicine.org

Requests from Editors:

Thank you for your attention to our initial requests. Please respond to the following before we proceed:

1) There are some instances of the term "subject." In Fig 1 title, please replace "subject" with participant, patient, individual, or person

2) In the discussion, please replace “compliance” with adherence

3) Please include line numbers in your next draft

Comments from Reviewers:

Reviewer #1: The authors have addressed all my points.

In response to an editorial request the authors have now adjusted for multiplicity. It is not clear to me what the logic behind this is nor exactly how it has been done. Has the adjustment been applied across all the analyses in the paper? That seems clearly wrong. Is it applied on the family of comparisons in a single table so in Table 2 the family size is 18 if I count them correctly. In that case why would you want to adjust each for all of them? Or has it been done separately for the set of comparisons per covariate. so for comorbid disease the family size is 6?

I think that at the very least the procedure needs to be clarified but more importantly it needs to be justified scientifically or clinically, and if not removed.

Michael Dewey

Reviewer #2: Dear authors,

Thanks for your detailed responses.

I have a few suggestions to improve your manuscript:

1. Table 1: line: Hypertension duration > 1 year, n (%) 0 (0.0%) 13268 (83.0%) 4406 (70.1%) 8862 (91.4%)

I think the percentages of remissions columns should refer to the 13268 subjects, like the entire table.

2. You found that males have more chance to reach remission. Can you speculate the reason because it occurs? Central obesity? Do you have data from abdominal circumference?

3. BMI is associated with a lower remission rate. It would be interesting if you could show a cutoff value. Is it possible?

Reviewer #3: Thank you for your thoughtful and thorough reply and appropriate revision in response to my initial review. This reviewer has no further comments or suggestions.

[LINK]

---

## [Editor Report · Decision Letter 3]

16 Sep 2021

Dear Dr Stenberg, 

On behalf of my colleagues and the Academic Editor, Dr. Sanjay Basu, I am pleased to inform you that we have agreed to publish your manuscript "Remission, relapse and risk of major cardiovascular events after metabolic surgery in persons with hypertension: a Swedish nationwide registry-based cohort study" (PMEDICINE-D-21-01171R3) in PLOS Medicine.

PRESS

Sincerely, 

Beryne Odeny 

Associate Editor 

PLOS Medicine